# Effectiveness of Natural-Based Coatings on Sweet Oranges Post-Harvest Life and Antioxidant Capacity of Obtained By-Products

**Deived Uilian de Carvalho** [1,2,3,*] , **Carmen Silvia Vieira Janeiro Neves** [2], **Maria Aparecida da Cruz** [1,2] , **Ronan Carlos Colombo** [4], **Fernando Alferez** [3] **and Rui Pereira Leite Junior** [1]

1 Área de Proteção de Plantas, Instituto de Desenvolvimento Rural do Paraná–IAPAR/Emater (IDR-Paraná), km 375 Celso Garcia Cid Road, Londrina 86047-902, Brazil; maria.cruz.bejatto@uel.br (M.A.d.C.); ruileite@idr.pr.gov.br (R.P.L.J.)
2 Centro de Ciências Agrárias, Universidade Estadual de Londrina (UEL), km 380 Celso Garcia Cid Road, Londrina 86057-970, Brazil; csvjneve@uel.br
3 Department of Horticulture, Southwest Florida Research and Education Center, University of Florida–Institute of Food and Agricultural Sciences (UF–IFAS), Immokalee, FL 34142, USA; alferez@ufl.edu
4 Centro de Ciências Agrárias, Universidade Federal Tecnológica do Paraná, Linha Santa Bárbara, Francisco Beltrão 85601-970, Brazil; ronancolombo@utfpr.edu.br
* Correspondence: deived.carvalho@fundecitrus.com.br

**Abstract:** The use of natural-based coatings is an eco-friendly approach that can be applied in citrus postharvest to preserve fruit quality and to prolong shelf life. Additionally, the study of antioxidant capacity of obtained by-products from fruits is of great value to mitigate better practices to manage the residues left from the juice processing industry. Under this context, the aim of this study was to investigate the use of carnauba wax/wood resin-based coating and cold storage on postharvest life of Valencia Late and Natal IAC sweet oranges, as well as the physicochemical quality and antioxidant capacity of its by-products. Mature fruits were harvested in 2019 and 2020 seasons. Initially, fruits were assessed for physicochemical quality and antioxidant capacity. Then, fruits were treated with carnauba wax and wood resin mixture and stored for 0, 15, 30, 45 and 60 days in a cold chamber. Fruit color index, weight loss, physicochemical quality and sensory profile of the fruits were monitored at harvest and during each cold storage period. Evaluations were performed in triplicates of 10-fruit. Valencia Late and Natal IAC fruits had proper quality in both years, attending the requirements of the fresh market and processing industry. Flavedo and albedo section displayed the highest concentration of bioactive compounds such as phenolics, flavonoids and antioxidant activity. The coating treatment associated with cold storage was efficient to preserve fruit color and to retard weight loss for both varieties up to 60 days. The sensory profile and quality of the carnauba wax/wood resin treated fruits were preserved all over the cold storage period, while uncoated fruits ranked low for most of the sensory attributes. Together, Valencia Late and Natal IAC fruits contain a high level of healthy beneficial compounds, which may be exploited as a natural source of low-cost antioxidants. Further, carnauba wax/wood resin coating associated with cold storage effectively reduce weight loss and color progression in sweet orange fruits, in addition to preserving overall physicochemical and sensory quality.

**Keywords:** *Citrus* ×*sinensis* (L.) Osbeck; late-season varieties; citrus by-products; cold storage; renewable fruit coatings

## 1. Introduction

Fresh fruit consumption increased worldwide, driven by consumers concerns on high-quality and healthy food habits [1]. *Citrus* spp. are widely cultivated, represented by a diversity of species grown in the tropical, subtropical and Mediterranean regions. Brazil

leads the sweet oranges [*Citrus ×sinensis* (L.) Osbeck] production. Nearly three-quarters of the sweet oranges are processed into frozen concentrated orange juice (FCOJ) and not-from-concentrate juice (NFC) [2]. The late-season varieties Valencia and Natal represent 25% and 10% of the Brazilian sweet orange orchards, respectively [3]. Even though these late-season varieties are mostly processed into juice, they can also be commercialized as fresh fruits.

The citrus-processing industry generates large amounts of by-products every year, including peel (flavedo and albedo), pulp (juice sacs), rag (segment wall and central core) and seed residues [4,5] that accounts around half of the total fruit weight [6]. Citrus by-products may become waste and a possible source of environmental pollution if not adequately processed [7,8]. These by-products are potent natural source for bioactive compounds as phenolics and antioxidants for human health [9]. Under this context, better practices on citrus by-products application are essential to determine their potential as healthy beneficial products, such as outstanding low-cost antioxidant sources [10].

Although attending the needs of the domestic fresh fruit market, the Brazilian citrus industry has a small share in the global market, mostly due to phytosanitary restrictions, associated with the lack of an appropriate postharvest processing structure. For instance, the citrus fruit can have a long postharvest life if appropriate practices are applied [1]. Proper postharvest practices for the citrus fresh fruit are essential to preserve quality, to extend shelf life and to avoid major losses. Fruit losses are substantially high in low-income countries due to the lack of postharvest management, involving limited refrigeration. In contrast, developed countries usually have good postharvest management, as the fruit undergo to a cold chain soon after harvest and remain under this condition until consumption [11]. The cold chain maintains the fresh fruit quality, as the cooling process retards the deterioration and losses associated with natural senescence due to a decrease in the metabolic rates [12]. However, investments in cold chain and high-tech postharvest management practices are still a challenge for low-income countries, where politics with efforts for nonhazardous and safe food management are taking place gradually.

Alternatively, the application of natural-based coating materials, including waxes and resins, on fresh fruit can be used by low-income countries where natural resources are available, recognized as eco-friendly approach [13]. These materials are becoming very popular in citrus postharvest, as they reduce losses by the differential permeability of $CO_2$, $O_2$ and water vapor, decreasing the metabolic rate and water losses [1,14]. Protective coatings may be extracted from different natural and renewable resources as carnauba palm [15,16], sugarcane [17], soybean [18], candelilla [19], coconut [20], wood [21,22] and beeswax [20,23]. Carnauba wax is derived from leaves of palm trees (*Copernica prunifera* (Mill.) HE Moore) which are native to the tropical rainforest of Brazil [24]. This wax is widely used in coating compounds to increase toughness and brightness in fruits [16,22]. Rosins, also known as wood resin, can also be applied as protective coatings on fresh fruit [21]. Rosins are residues left after distillation of the volatile fraction of pine oil and turpentine from the crude resin of the pine trees [21,22]. Together, carnauba wax and wood resin are allowed as non-synthetic ingredients in coatings mixtures for organic citrus production [1].

The application of renewable coating materials on fresh citrus is an effective postharvest practice to preserve quality and to prolong shelf life by improving visual aspect and reducing postharvest losses [14]. Furthermore, the knowledge about antioxidant capacity of distinct fruit tissues of the most cultivated varieties, particularly those that are widely used by the processing industry, is of great value in order to mitigate better practices to manage the residues left from the juice processors contributing to the sustainability of the citrus industry. Accordingly, we report in this paper the physicochemical quality and the antioxidant capacity of Valencia Late and Natal IAC sweet oranges fruits and their by-products, the most exploited late-season varieties in Brazil, as well as the effectiveness of the carnauba wax and wood resin-based coatings on fruit postharvest shelf life under cold storage.

## 2. Materials and Methods

### 2.1. Fruit Harvest and Treatments

Mature fruits of Valencia Late and Natal IAC sweet oranges were harvested in mid-November 2019 for fruit characterization and mid-November 2020 for postharvest assay from an experimental orchard located in Guairaçá, state of Paraná, Brazil. The sweet orange trees were grafted on Rangpur lime (*C.* × *limonia* (L.) Osbeck) and were seven to eight years old. Three replications of 20 fruits per variety were collected from 10 trees in 2019, while 500 fruits per variety were harvested in 2020. After harvest, fruits were immediately transferred to a cold chamber for pre-cooling for 12 h at $5 \pm 1$ °C and 60–70% RH at the facility of the Instituto de Desenvolvimento Rural do Paraná–IAPAR/Emater (IDR-Paraná) in Londrina, state of Paraná, Brazil. Fruits were hand-washed with neutral detergent under tap water and submerged in 1.0% NaCl solution. For the postharvest assay, half of each fruit batch (250 fruit) was coated with carnauba wax/wood resin (rosin, 10:1) at 18% total solids (Aruá, BR 18, Matão, SP, Brazil). The other half of the fruits was uncoated and used as control. Fruits were packed in industrial plastic boxes (40.8 kg capacity) and stored for 0, 15, 30, 45 and 60 days in a cold chamber at $5 \pm 1$ °C and 60–70% RH.

### 2.2. Pre-Storage Fruit Quality Evaluation

Three replicates of 10 fruits per variety were evaluated in regard to physical characteristics, i.e., weight, length, diameter, shape index, flavedo and albedo thickness, color, juice content and number of seeds; and chemical attributes, i.e., total soluble solids (TSS), titratable acidity (TA) and TSS·TA$^{-1}$ ratio. Total phenolics, total flavonoids and the percentage of DPPH (2,2-diphenyl-1-picrylhydrazyl) scavenging in different fruit tissues were determined using the other three replicates of 10 fruits per variety.

Fruit samples were weighed (g) using a semi-analytical scale (Bel, M505, Piracicaba, SP, Brazil). Fruit length, diameter, flavedo and albedo thickness (mm) were measured with a digital Vernier caliper (Mitutoyo, ABS, Kawasaki, Kanagawa, Japan) and classified according to the fresh fruit standard [25]. Fruit shape index was calculated based on the relation between fruit length and diameter.

Fruit color was determined by three readings along the equatorial circumference of each fruit using a digital colorimeter (Konica Minolta Sensing Americas Inc., CR-400, Ramsey, MN, USA). This attribute was expressed as citrus color index (CCI) based on the CIE L*a*b* color space. The CCI is a comprehensive indicator of color impression with positive values for red, negative values for blue-green and zero for an intermediate mixture of red, yellow and blue-green [26]. Fruit samples were juiced using a Croydon extractor (Croydon, ES4EA-B60000, Duque de Caxias, RJ, Brazil). Juice content was determined according to the following equation:

$$\text{JC} = \frac{\text{JW}}{\text{FW}} \times 100, \tag{1}$$

where JC = juice content (%), FW = fruit weight (g) and JW = juice weight (g).

The number of seeds was manually counted. TSS was measured using an aliquot of undiluted juice (0.3 mL) in a digital refractometer (Atago Co., Ltd., PAL-3, Tokyo, Kantō, Japan) at 20 °C. The TSS was expressed in °Brix units. Titratable acidity was determined in 25 mL of diluted juice (juice: distilled water; 1:4), 0.1 N NaOH and phenolphthalein as indicator in a TitroLine easy titrator (Schott Instruments GmbH, TitroLine easy, Mainz, Rhineland-Palatinate, Germany). The results were expressed in grams of citric acid per 100 mL of juice (g·100 mL$^{-1}$) [27]. The ratio (TSS·TA$^{-1}$) was then calculated.

The fruit tissues were manually separated into flavedo, albedo, juice sacs, segment wall and central core (Figure 1). Then, they were freeze-dried at –40 °C under vacuum (Labconco Co., Freeze Drier 8, Kansas City, KA, USA) until total dehydration (~50 h). Before analyses, the fruit tissues were ground in liquid nitrogen using a mortar and pestle. Fifty milligrams of powdered tissue were extracted twice with 1.0 mL of absolute methanol for 30 min in the dark, at room temperature. After centrifugation at 20,000× *g* (Centrifuge

5417 C, Eppendorf, Hamburg, Germany) for 5 min at 4 °C, supernatants were filtered through a 0.45 μm syringe filter (Corning Inc., SFCA, Tewksbury, MA, USA) to obtain a clear solution which was dried in vacuum (Concentrator Plus/Vacufuge® Plus, Eppendorf, Hamburg, Germany) for 1.5 h at room temperature. The dried extracts were re-suspended in 1.0 mL Milli-Q water, vortexed and centrifuged at 20,000× *g* (Centrifuge 5417 C, Eppendorf, Hamburg, Germany) for 1 min at 4 °C. The supernatants were used to quantify total phenolics, total flavonoids and the percentage of DPPH scavenging by spectrophotometry (Molecular Devices, Spectramax 190, San Jose, CA, USA). All assays were conducted in triplicate.

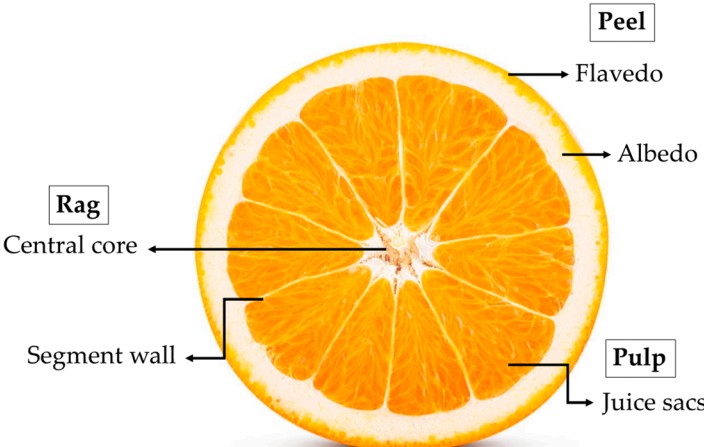

**Figure 1.** Scheme of fruit cross section showing different tissues extracted from Valencia Late and Natal IAC sweet orange, including peel (flavedo and albedo), pulp (juice sacs) and rag (central core and segment wall).

Total phenolic content was determined according to Rahman et al. [28]. An aliquot of 12.5 μL of the supernatant was diluted in 50 μL Milli-Q water and maintained at room temperature for 1 h with 12.5 μL of 1 N Folin-Ciocalteau and 125 μL of 7% $NaCO_3$. The absorbance was measured against a blank at 750 nm in a spectrometer (Molecular Devices, Spectramax 190, San Jose, CA, USA) and the total phenolic content was determined by using a standard calibration curve developed with gallic acid (Sigma-Aldrich, Saint Louis, MO, USA). Phenolic concentrations were expressed as gallic acid equivalent (mg GAE × 100 g dry tissue$^{-1}$).

The total flavonoids was determined using the methodology described by Wang et al. [7] with some modifications. An aliquot of 25 μL of the supernatant was mixed with 25 μL of 5% $NaNO_2$ and incubated for 6 min at room temperature in the dark. After that, 25 μL of 10% $AlCl_3$ was added to the reaction and incubated again for 6 min. The reaction was stopped by adding 125 μL of 4% NaOH. After 15 min in the dark at room temperature for color development, the absorbance was measured at 350 nm in a spectrometer (Molecular Devices, Spectramax 190, San Jose, CA, USA). Total flavonoids was expressed as hesperidin equivalents (mg of hesperidin × 100 g dry tissue$^{-1}$) through the calibration curve developed with hesperidin (Sigma-Aldrich, Saint Louis, MO, USA).

The 2,2-diphenyl-1-picrylhydrazyl (DPPH) radical scavenging activity of the fruit tissues was measured using the method previously reported by He et al. [29], with minor modifications. An aliquot of 150 μL 0.2 mM DPPH radical diluted in methanol was mixed with 50 μL of the supernatant. Then, the mixture was vortexed for 30 s and incubated for 30 min in the dark at room temperature for color development. After incubation, the absorbance of the reaction progress was recorded at 517 nm in a spectrometer (Molecular

Devices, Spectramax 190, San Jose, CA, USA). The percentage of inhibition of DPPH radical was expressed and calculated according to He et al. [29]:

$$\% \text{ Antioxidant activity} = \frac{\text{Abs}_{\text{control}} - \text{Abs}_{\text{sample}}}{\text{Abs}_{\text{control}}} \times 100, \quad (2)$$

where $\text{Abs}_{\text{control}}$ = absorbance of the control (only with solvent); and $\text{Abs}_{\text{sample}}$ = absorbance of the sample.

### 2.3. Fruit Postharvest Quality Assay

Carnauba wax/wood resin coated, and uncoated fruits were evaluated periodically at 0, 15, 30, 45 and 60 days during cold storage for the following characteristics: fruit color index, weight loss, juice content, TSS, TA, ratio and sensory profile. A triplicate of 10 fruits per treatment was evaluated each time. Fruit color, juice content, TSS, TA and ratio were determined as described previously. Fruits were weighed at harvest and the weight loss was monitored after each period of cold storage using a semi-analytical scale (Bel, M505, Piracicaba, SP, Brazil). Total weight loss was expressed as a percentage according to the standard method of the AOAC [27].

The sensory profile of the fruits was assessed by 12-trained panelists, five men and seven women, after each period of cold storage and wax coating treatment. Participants were asked voluntarily to take part of the sensory tasting sessions, without any compensation. Panelists were frequent sweet orange consumers, and they were recruited from a pool of staff and graduate students, with ages ranging from 17 to 53 years old, from the IDR-Paraná, Londrina, state of Paraná, Brazil. The evaluations were performed in a standard testing room with white LED lights. Ten fruits per treatment were assessed during each session, at harvest and 15, 30, 45 and 60 days after cold storage. Prior to the sensory evaluations, fruit samples were equilibrated at room temperature for 1 h and were then hand-peeled. Fruits were separated in segments and sectioned into uniform pieces. Each panelist received a whole fruit and three peeled sectioned fruit segments prepared from different fruit for each treatment. The samples were placed on white disposable plate randomly arranged and identified by three-digit random codes (triads) to avoid potential biases during the sessions. Panelists were provided with a glass of water at room temperature for palate cleansing and instructed to rinse their mouth with water before the session and between samples. After signing the consent form of the study, the panelists were instructed to rate the sensory attributes using a nine-point hedonic scale. Based on their preference, the panelists rated the following sensory attributes for the fruit: color, texture, firmness, aroma, flavor, juiciness, and overall preference. The scores rated by the panelists were used to build the preference mapping to describe the sensory attributes.

### 2.4. Statistical Assessment

The statistical design for both fruit characterization and postharvest assay was completely randomized. The data were tested for normal distribution and homogeneity at $p \leq 0.05$. One-way analysis of variance (ANOVA) was used to determine significant variation between the two tested sweet orange varieties for each physicochemical attribute. Differences were compared by Student's t test ($p \leq 0.05$). For the antioxidant capacity and postharvest assay, the effect of the two tested factors and the interaction between them were analyzed by two-way ANOVA. All differences were considered significant at $p \leq 0.05$, according to the Tukey's test. The experimental design for the antioxidant capacity was arranged in a factorial scheme with two sweet orange varieties (Valencia Late and Natal IAC) and five fruit section (flavedo, albedo, central core, juice sacs and segment wall). While a factorial scheme with four treatments (two varieties × two coating treatments) and five storage periods (0, 15, 30, 45 and 60 days of cold storage) was designed for the postharvest assay. All analyses were performed using the software program R v. 4.0.2 (The R Foundation for Statistical Computing, Vienna, Austria).



## 3. Results

### 3.1. Pre-Storage Fruit Quality Evaluations

Fruits of Valencia Late and Natal IAC were initially characterized according to their physicochemical quality attributes (Table 1) and antioxidant capacity (Table 2). Valencia Late fruits were smaller and lighter than the ones of Natal IAC (Table 1). The fruit shape indices ranged from 0.93 for Valencia Late to 0.99 for Natal IAC, indicating a round or sub-globose shape. The citrus color index (CCI) was relatively higher for Natal IAC (2.11) than Valencia Late (1.57). The two tested varieties showed similarities for flavedo and albedo thickness, as well as for the number of seeds per fruit. The lowest juice content in fruit was for Valencia Late with 36%, while the fruits of Natal IAC had higher juice content, >41%. Furthermore, Natal IAC fruits had higher total soluble solids (TSS) and titratable acidity (TA). Similar sugar-acidity ratios (TSS·TA$^{-1}$) were found in the juice of both varieties, ranging from 11.2 for Natal IAC up to 12.0 for Valencia Late.

**Table 1.** Physicochemical quality of Valencia Late and Natal IAC sweet orange fruits for the 2019 harvest. Each value represents the mean of triplicate samples (mean value ± standard deviation).

| Fruit Attribute | Valencia Late | Natal IAC | CV (%) [1] | *F* Value |
|---|---|---|---|---|
| Fruit length—FL (mm) | 70.0 ± 1.17 [a][2] | 72.7 ± 2.95 [a] | 3.15 | 2.21 [ns] |
| Fruit diameter—FD (mm) | 75.0 ± 1.18 [a] | 73.4 ± 0.64 [a] | 1.28 | 4.10 [ns] |
| Fruit shape index (FL·FD$^{-1}$) | 0.93 ± 0.01 [a] | 0.99 ± 0.05 [a] | 3.49 | 4.36 [ns] |
| Citrus color index—CCI | 1.57 ± 0.34 [b] | 2.11 ± 0.06 [a] | 13.0 | 7.60[*] |
| Flavedo thickness (mm) | 2.68 ± 0.30 [a] | 2.74 ± 0.15 [a] | 8.77 | 0.09 [ns] |
| Albedo thickness (mm) | 2.36 ± 0.25 [a] | 2.33 ± 0.11 [a] | 8.24 | 0.05 [ns] |
| Fruit weight (g) | 162 ± 2.25 [a] | 150 ± 8.67 [b] | 3.99 | 11.8 [*] |
| Number of seeds | 2 ± 0.92 [a] | 2 ± 0.40 [a] | 36.0 | 0.21 [ns] |
| Juice content (%) | 36.1 ± 2.91 [a] | 41.3 ± 2.39 [a] | 6.88 | 5.86 [ns] |
| Soluble solids content—SST (°Brix) | 11.2 ± 0.95 [a] | 12.1 ± 0.64 [a] | 6.95 | 1.98 [ns] |
| Titratable acidity—TA (g·100 mL$^{-1}$) | 0.93 ± 0.13 [a] | 1.08 ± 0.03 [a] | 9.49 | 3.52 [ns] |
| Ratio (SST·TA$^{-1}$) | 12.0 ± 0.65 [a] | 11.2 ± 0.88 [a] | 6.62 | 1.61 [ns] |

[1] CV, coefficient of variation; [2] means followed by the same letter in the row do not significantly differ according to Student's *t* test. Significance level: ns, non-significant; *, $p \leq 0.05$.

**Table 2.** Total phenolic content (mg GAE·100 g of dry tissue$^{-1}$), total flavonoid content (mg hesperidin per 100 g of dry tissue), and percentage of DPPH (2,2-diphenyl-1-picrylhydrazyl) scavenging activity of different fruit tissues (flavedo, albedo, central core, juice sacs and segment wall) of Valencia Late and Natal IAC sweet orange fruits. Each value represents the mean of triplicate samples (mean value ± standard deviation).

| Source of Variance | Total Phenolics (mg 100g$^{-1}$) | | |
|---|---|---|---|
| | Valencia Late | Natal IAC | Mean |
| Flavedo | 246.9 ± 4.1 [Ba][1] | 267.9 ± 4.4 [Aa] | 257.4 ± 4.3 |
| Albedo | 200.0 ± 5.4 [Bb] | 227.1 ± 6.0 [Ab] | 213.6 ± 5.7 |
| Central core | 161.3 ± 5.4 [Ac] | 166.0 ± 7.5 [Ac] | 163.6 ± 6.5 |
| Juice sacs | 107.7 ± 5.1 [Ad] | 117.5 ± 6.0 [Ad] | 112.6 ± 5.6 |
| Segment wall | 85.9 ± 3.9 [Ae] | 82.7 ± 8.7 [Ae] | 84.3 ± 6.3 |
| Mean | 160.3 ± 4.8 | 172.3 ± 6.5 | |
| CV (%) [2] | 3.53 | | |
| Fruit section | 876.6 *** | | |
| Variety | 30.8 *** | | |
| Fruit section × Variety | 6.48 *** | | |

**Table 2.** *Cont.*

|  | Total flavonoids (mg 100g$^{-1}$) | | |
|---|---|---|---|
|  | Valencia Late | Natal IAC | Mean |
| Flavedo | 235.5 ± 18.5 [Aa] | 241.0 ± 3.6 [Aa] | 238.6 ± 11.1 |
| Albedo | 167.2 ± 13.2 [Bb] | 221.9 ± 10.6 [Ab] | 194.6 ± 18.4 |
| Central core | 129.7 ± 10.2 [Ac] | 137.6 ± 9.6 [Ac] | 133.7 ± 9.9 |
| Juice sacs | 38.1 ± 3.7 [Bd] | 60.2 ± 8.2 [Ad] | 49.1 ± 6.0 |
| Segment wall | 23.9 ± 1.7 [Bd] | 51.3 ± 9.6 [Ad] | 37.6 ± 5.7 |
| Mean | 118.9 ± 9.5 | 142.4 ± 8.3 |  |
| CV (%) | 7.72 |  |  |
| Fruit section | 456.3 *** |  |  |
| Variety | 40.7 *** |  |  |
| Fruit section × Variety | 5.75 ** |  |  |
|  | DPPH scavenging (%) | | |
|  | Valencia Late | Natal IAC | Mean |
| Flavedo | 74.1 ± 3.8 | 74.8 ± 3.6 | 74.4 ± 3.7 [a] |
| Albedo | 66.5 ± 2.7 | 64.7 ± 1.6 | 65.6 ± 2.2 [b] |
| Central core | 60.7 ± 3.3 | 54.9 ± 4.8 | 57.8 ± 4.1 [c] |
| Juice sacs | 51.6 ± 0.5 | 41.4 ± 3.8 | 46.5 ± 2.2 [d] |
| Segment wall | 37.7 ± 1.4 | 33.2 ± 2.5 | 35.4 ± 2.0 [e] |
| Mean | 58.1 ± 2.3 [A] | 53.8 ± 3.3 [B] |  |
| CV (%) | 5.55 |  |  |
| Fruit section | 147.3 *** |  |  |
| Variety | 14.4 ** |  |  |
| Fruit section × Variety | 2.64 [ns] |  |  |

[1] Means followed by the same letter, lowercase in the column and capital case in the row, do not significantly differ according to the Tukey´s test. [2] CV, coefficient of variation. Significance level: ns, non-significant; **, $p \leq 0.01$; ***, $p \leq 0.001$.

The total of phenolics and flavonoids as well as the percentage of DPPH scavenging activity in the different fruit tissues including flavedo, albedo, central core, juice sacs and segment wall of both sweet oranges were assessed (Table 2). The highest content of phenolics and flavonoids as well as the DPPH scavenging activity were observed in the flavedo tissues of the fruits of both sweet orange varieties, while the juice sacs and the segment wall showed the lowest values. The albedo and central core tissues had intermediate values for these compounds. Regarding the phenolic content, Natal IAC fruits scored ($p \leq 0.05$) higher level in flavedo (268 ± 4.4 mg GAE·100 g$^{-1}$) and albedo (227 ± 6.0 mg GAE·100 g$^{-1}$) tissues than Valencia Late fruits (247 ± 4.1 and 200 ± 5.4 mg GAE·100 g$^{-1}$, respectively). However, no such differences were observed for all other fruit tissues. Significant differences ($p \leq 0.05$) were also found for the total flavonoids, as Natal IAC fruits had higher levels in the albedo, juice sacs and segment wall tissues (222 ± 10.8, 60.2 ± 8.2, 51 ± 9.6 mg of hesperidin per 100 g, respectively) than Valencia IAC (167 ± 13.2, 38.0 ± 3.7, 24 ± 1.7 mg of hesperidin per 100 g, respectively). In regard to the DPPH scavenging activity, the juice sacs of the Valencia Late fruits had significantly higher ($p \leq 0.05$) antioxidant capacity (52 ± 0.5% of DPPH scavenging) than the Natal IAC juice sacs (41 ± 3.8% of DPPH scavenging).

### 3.2. Postharvest Fruit Quality Evaluations

#### 3.2.1. Weight Loss

The weight loss of the fruits of the two sweet orange varieties, Valencia Late and Natal IAC, increased significantly ($p \leq 0.001$) over the evaluated period, particularly for those without wax coating treatment (Figure 2). Fruits of both sweet orange varieties were similar for the same wax-coating treatment at each period of storage, but had differences in regard to the postharvest treatments, wax coating or non-coating. After 60 days of cold storage, fruits without coating treatment exceeded 25% of weight loss for both varieties.



On the other hand, the wax-coating fruit had a loss rate below 15%. For all four-storage periods, the non-coating fruits had twice the weight loss compared to the wax-coating ones, supporting the effectiveness of this postharvest treatment.

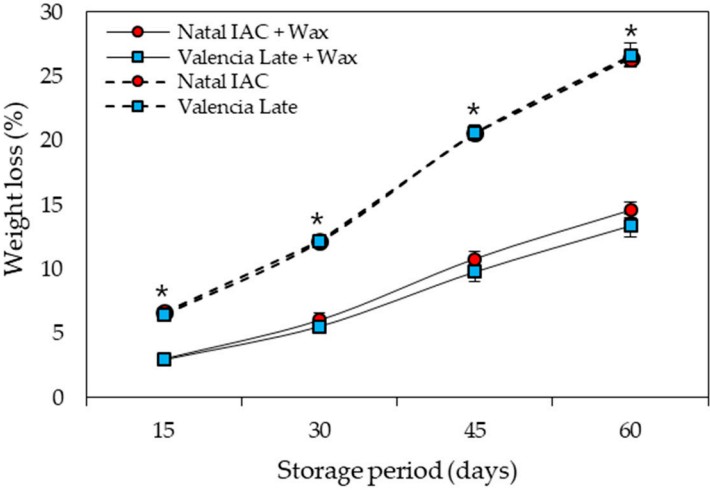

**Figure 2.** Weight loss (%) of Valencia Late and Natal IAC sweet orange fruits treated with carnauba wax/wood resin and maintained under different cold storage periods (15, 30, 45 and 60 days). Significance level: *, $p \leq 0.05$.

### 3.2.2. Fruit Color Index

The color development of the fruits of the two assessed sweet oranges were monitored for the entire storage period, from 0 (harvest) up to 15, 30, 45 and 60 days after harvest (Figure 3). As main effects, the storage period and variety were highly significant ($p \leq 0.001$), but no significant interaction of these two factors was found in the postharvest assay (Figure 3). The fruit color index progressed during the storage period for the uncoated treatment (control) in both varieties. In contrast, no color progression was observed for the wax-coating treatment, regardless of the sweet orange varieties. Differences in the wax-coating treatments were observed after 30 days of cold storage but become evident ($p \leq 0.05$) at 45 and 60 days of storage. Fruits of both varieties had similar performance for color development.

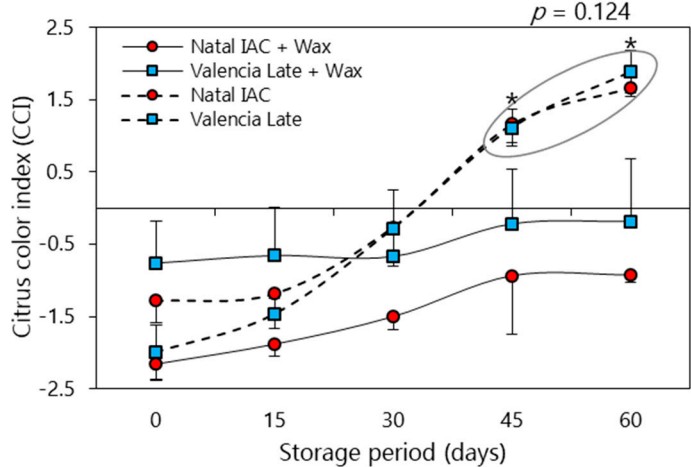

**Figure 3.** Citrus color index (CCI) of Valencia Late and Natal IAC sweet orange fruits subjected to carnauba wax/wood resin coating treatment after 0 (harvest), 15, 30, 45 and 60 days of cold storage. Significance level: *, $p \leq 0.05$.

### 3.2.3. Juice Quality

The juice quality of the Valencia Late and Natal IAC fruits that received the uncoated and wax-coating postharvest treatments was evaluated at 0 (harvest), 15, 30, 45 and 60 days after cold storage (Table 3). No significant interactions ($p \leq 0.05$) were found between the postharvest treatment and storage period. In contrast, significant differences ($p \leq 0.05$) were observed for TSS and TSS.TA$^{-1}$ ratio for the postharvest treatments. The highest TSS content was found in wax-coated and uncoated fruit of Valencia Late and for uncoated Natal IAC fruits. Valencia Late fruits also displayed high TSS.TA$^{-1}$ ratio with an average of 18.7, significantly different from the wax-coated fruits of Natal IAC. The acidity level was also similar among the treatments, ranging from 0.62 up to 0.75 g of citric acid per 100 mL of juice. No significant differences were found among the treatments across the evaluation period.

**Table 3.** Total soluble solids (TSS), titratable acidity (TA) and TSS/TA$^{-1}$ ratio of Valencia Late and Natal IAC sweet orange fruits treated and non-treated with carnauba wax/wood resin coating and maintained at different cold storage periods (mean value ± standard deviation).

| Source of Variance | Total Soluble Solids—TSS (°Brix) | | | | |
| --- | --- | --- | --- | --- | --- |
| | Valencia Late + Wax | Natal IAC + Wax | Valencia Late | Natal IAC | Mean |
| 0 day (harvest) | 10.7 ± 0.70 | 10.6 ± 0.62 | 11.1 ± 0.43 | 11.7 ± 1.47 | 11.1 ± 0.81 |
| 15 days | 11.5 ± 0.32 | 10.6 ± 0.58 | 11.9 ± 0.50 | 11.9 ± 1.41 | 11.5 ± 0.70 |
| 30 days | 11.7 ± 1.38 | 10.5 ± 0.40 | 12.2 ± 1.65 | 11.2 ± 0.55 | 11.4 ± 1.00 |
| 45 days | 12.0 ± 0.50 | 10.0 ± 0.26 | 12.7 ± 0.40 | 11.8 ± 0.55 | 11.6 ± 0.43 |
| 60 days | 11.5 ± 0.10 | 10.1 ± 0.66 | 12.6 ± 1.70 | 11.4 ± 1.11 | 11.4 ± 0.89 |
| Mean | 11.5 ± 0.60 [A][1] | 10.4 ± 0.50 [B] | 12.1 ± 0.94 [A] | 11.6 ± 1.02 [A] | |
| CV (%) [2] | 7.97 | | | | |
| Storage period | 0.64 [ns] | | | | |
| Treatment | 9.97 [***] | | | | |
| Storage period × Treatment | 0.73 [ns] | | | | |
| | Titratable acidity—TA (g.100 mL$^{-1}$) | | | | |
| | Valencia Late + Wax | Natal IAC + Wax | Valencia Late | Natal IAC | Mean |
| 0 day (harvest) | 0.54 ± 0.05 | 0.68 ± 0.15 | 0.69 ± 0.15 | 0.67 ± 0.11 | 0.64 ± 0.12 |
| 15 days | 0.64 ± 0.11 | 0.67 ± 0.14 | 0.71 ± 0.17 | 0.78 ± 0.22 | 0.70 ± 0.16 |
| 30 days | 0.63 ± 0.09 | 0.65 ± 0.10 | 0.77 ± 0.22 | 0.71 ± 0.13 | 0.69 ± 0.14 |
| 45 days | 0.65 ± 0.03 | 0.77 ± 0.04 | 0.74 ± 0.19 | 0.77 ± 0.09 | 0.73 ± 0.09 |
| 60 days | 0.65 ± 0.04 | 0.79 ± 0.07 | 0.82 ± 0.08 | 0.79 ± 0.22 | 0.76 ± 0.10 |
| Mean | 0.62 ± 0.06 | 0.71 ± 0.10 | 0.75 ± 0.16 | 0.74 ± 0.15 | |
| CV (%) | 19.60 | | | | |
| Storage period | 1.21 [ns] | | | | |
| Treatment | 2.74 [ns] | | | | |
| Storage period × Treatment | 0.18 [ns] | | | | |
| | TSS.TA$^{-1}$ratio | | | | |
| | Valencia Late + Wax | Natal IAC + Wax | Valencia Late | Natal IAC | Mean |
| 0 day (harvest) | 20.1 ± 1.88 | 16.3 ± 5.06 | 16.5 ± 3.36 | 17.7 ± 2.74 | 17.7 ± 3.26 |
| 15 days | 18.3 ± 3.09 | 16.3 ± 4.55 | 17.5 ± 4.02 | 16.2 ± 5.13 | 17.1 ± 4.19 |
| 30 days | 18.6 ± 1.14 | 16.3 ± 2.22 | 16.4 ± 2.78 | 16.2 ± 2.67 | 16.9 ± 2.20 |
| 45 days | 18.6 ± 1.12 | 13.0 ± 0.68 | 18.0 ± 4.69 | 15.7 ± 1.22 | 16.2 ± 1.92 |
| 60 days | 17.8 ± 1.30 | 13.0 ± 2.03 | 15.4 ± 1.86 | 15.4 ± 6.68 | 15.5 ± 2.96 |
| Mean | 18.7 ± 1.71 [A] | 15.0 ± 2.91 [B] | 16.8 ± 3.34 [AB] | 16.2 ± 3.67 [AB] | |
| CV (%) | 18.89 | | | | |
| Storage period | 0.77 [ns] | | | | |
| Treatment | 3.24 [*] | | | | |
| Storage period × Treatment | 0.28 [ns] | | | | |

[1] Means followed by the same letter in the row do not significantly differ according to the Tukey´s test. [2] CV, coefficient of variation. Significance level: ns, non-significant; *, $p \leq 0.05$; ***, $p \leq 0.00$.

In contrast, we found a highly significant ($p \leq 0.001$) interaction between the postharvest treatment and cold storage period for juice content (Figure 4). Juice content fluctuated from 40% at harvest for fruits of both varieties that were wax-coated down to the lowest level of 25% at the end of the cold storage period for the Valencia Late fruits without any coating. Fruits of both varieties showed a progressive decrease in juice content throughout the storage period. However, no such tendency was observed for fruit of Natal IAC treated with carnauba wax and wood resin.

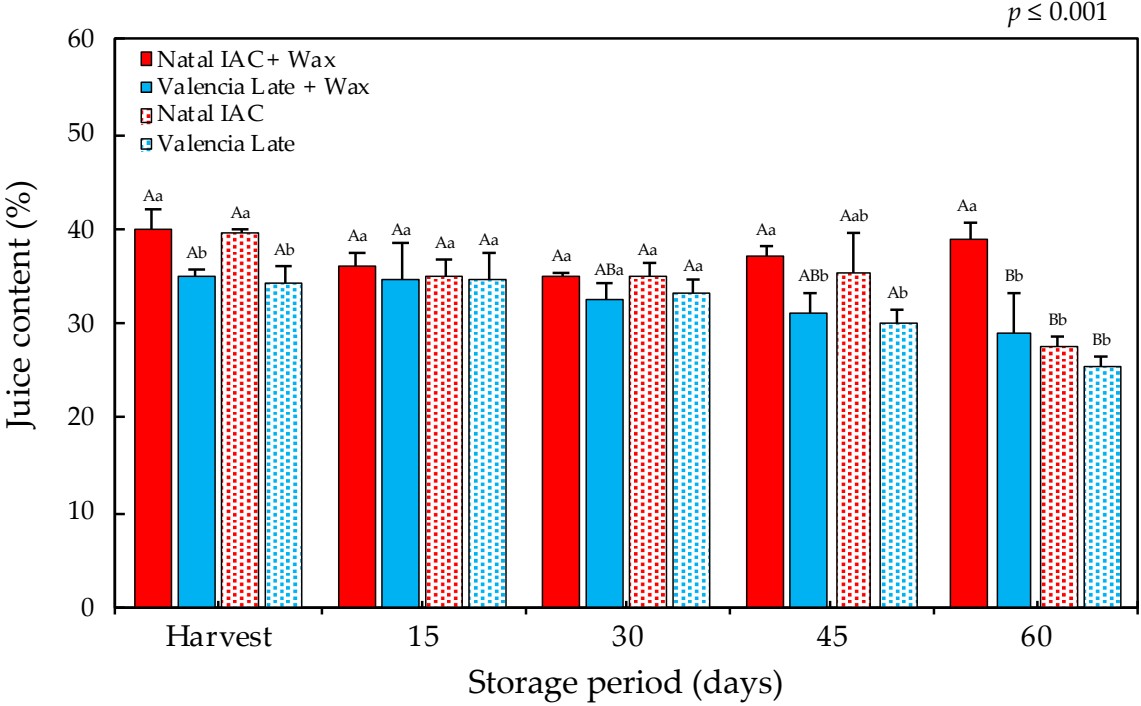

**Figure 4.** Juice content of Valencia Late and Natal IAC sweet orange fruits treated and non-treated with carnauba wax/wood resin coating after 0 (harvest), 15, 30, 45 and 60 days of cold storage. Bars followed by the same capital or lowercase letter do not differ significantly in regard to the storage period or coating treatment, respectively, according to the Tukey's test.

### 3.2.4. Sensory Analysis

Based on a nine-point hedonic scale, the preference mapping of the sensory attributes was described for the postharvest treatments and varieties throughout the storage period (Figure 5). Wax-coated fruits were more color-preferred by the panelists than uncoated fruits from the beginning of the postharvest assay, at harvest, up to 30 days of cold storage. However, no such tendency was observed after 30 days of cold storage when uncoated (control) fruits had an improvement in color development and, consequently, preference by the panelists. Fruit firmness and texture scores were gradually reduced throughout the storage period. In contrast, no substantial differences were observed in aroma and taste of the fruits due to the postharvest treatments in most of the evaluated periods, except at 60 days of cold storage. At that period, uncoated fruits were mostly disliked by the panelist pool compared to wax-coated fruit. Regarding fruit juiciness, no variation was perceived by the panelists. Based on the overall preference, both varieties and postharvest treatments did not differ at harvest, 15, 30 and 45 days of cold storage, but differences were reported after 60 days, when the Valencia Late fruits were most preferred by the panelists.

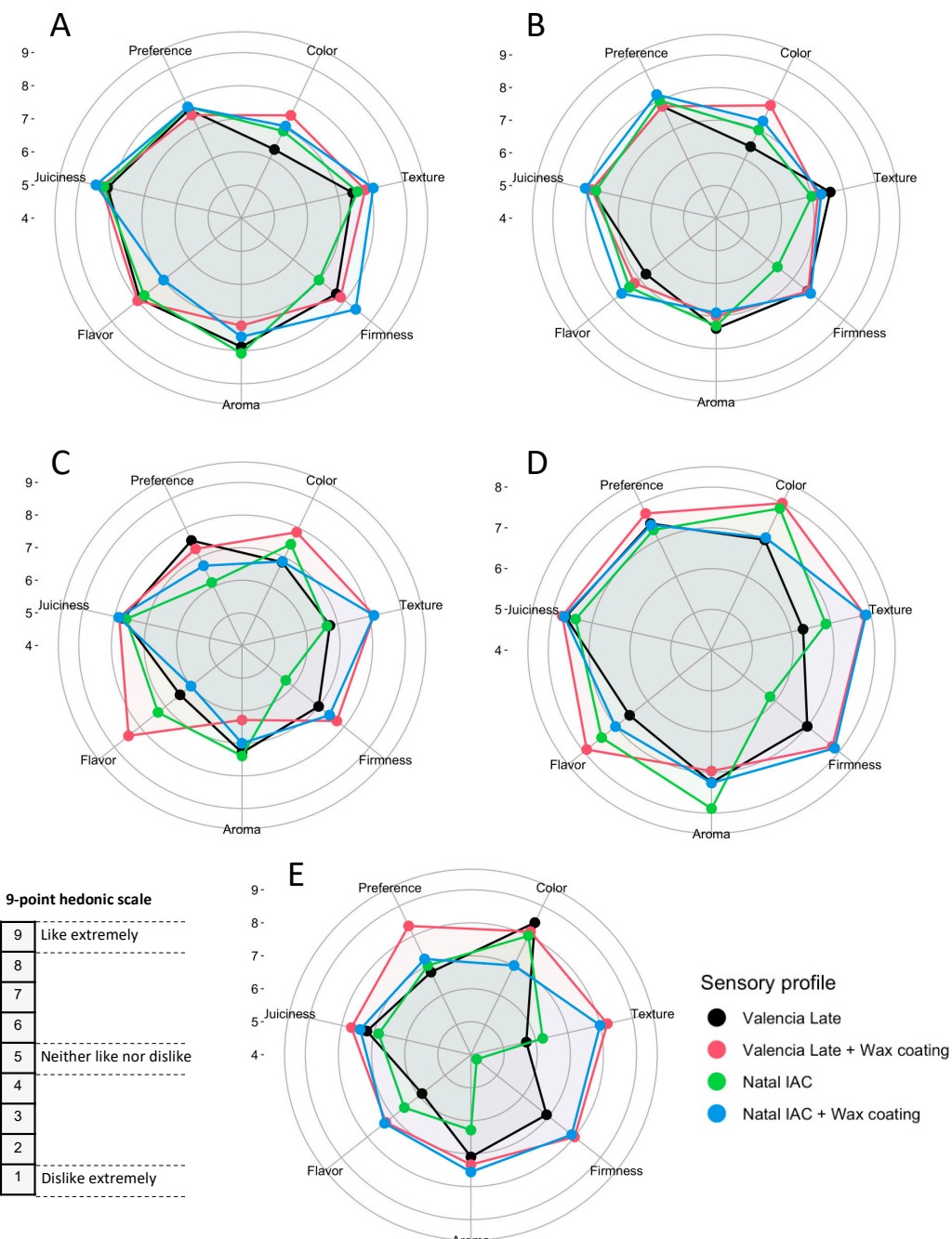

**Figure 5.** Sensory evaluation for color, texture, firmness, aroma, flavor, juiciness and preference of Valencia Late and Natal IAC sweet orange fruits treated and non-treated with carnauba wax/wood resin coating at harvest (**A**) and after 15 (**B**), 30 (**C**), 45 (**D**) and 60 days (**E**) of cold storage.

## 4. Discussion

### 4.1. Pre-Storage Fruit Quality Evaluations

Based on fruit characterization, Valencia Late and Natal IAC fruits were considered as large-sized (>71 mm) with an optimal grade (A) according to the Brazilian standards for citrus fresh fruit [25]. Fruits of both varieties were nearly round, as the shape indices were very close to 1.0 that indicates a sub-globose or round shape [30]. Natal IAC fruits exhibited a deeper yellowish-orange coloration (2.11 CCI) compared to Valencia Late ones (1.57 CCI).

The thickness of the flavedo and albedo, i.e., fruit rind or peel, was similar and thin (<3.0 mm) for fruits of both varieties (Table 1). There is a wide diversity in thickness of the citrus rind depending on scion–rootstock pairing and soil and climatic conditions [31]. Indeed, fruit rind plays an important role in maintaining fruit quality, as it regulates the water exchange between fruit and environment. Citrus fruit produced in cooler climates and higher altitudes usually develop a compact and thin-skinned peel, whereas those from warmer humid climate have a thicker rind [1]. Based on these morphological traits, the postharvest techniques involving coatings materials on fruit surface are imperative to preserve fruit quality and to extend shelf life by reducing water loss and shrinkage [32,33]. Moreover, the rind of citrus fruit from thinner albedo has a faster water potential adjustment compared to the one from thicker albedo [34], which may reduce problems of physiological disorders such as the postharvest peel pitting [35].

The Valencia Late and Natal IAC fruit can be considered commercially seedless, as they contained less than eight seeds per fruit [36]. Both varieties produced fruit with juice of excellent quality. The TSS contents were above the minimal standard [25,37] and within the values described by Pozzan and Triboni [38] for these varieties. The levels of citric acid were close to the range considered optimal for fresh fruit consumption, 0.5 to 1.0 g·100 mL$^{-1}$ [39]. Similarly, the sugar/acidity ratio range was substantially higher than that required by the international fresh citrus market, 6.5:1 [37].

Natal IAC fruits showed significatively ($p \leq 0.05$) higher phenolics concentration than Valencia Late for the albedo and flavedo tissues, as reported by Wang et al. [7]. Phenolics may play a role as antioxidant and antifeedants compounds, contributing to plant pigmentation, and working as attractants to pollinators, and protective agents against UV light, pathogens, herbivorous animals and predators [40]. According to Barros et al. [41], flavedo and albedo contain high levels of phenolics because they are in the outer part of the fruit and more predisposed to the synthesis of these compounds due to external stresses. Fruit tissues with higher phenolic contents generally show stronger antioxidant capacity [42], as confirmed in our study. However, the amount of total phenolics varied among the citrus varieties. Ramful et al. [43], assessing the flavedo extracts of 21 varieties of citrus, reported a broad spectrum in the phenolic concentration, which ranged from 188 ± 6.5 up to 767 ± 5.7 mg GAE·100 g$^{-1}$, depending on the species.

Flavonoids are among the major groups of phenolic compounds present in citrus fruits [7]. These phytochemicals help to protect plants against external stresses [44]. The concentration of flavonoids ranged from 32.7 ± 1.0 to 49.2 ± 1.3 mg of rutin g$^{-1}$ in the peel of eight citrus varieties cultivated in Taiwan [7]. In our study, the highest concentration of total flavonoids was found in the flavedo and albedo tissues but with a higher content in the albedo of Natal IAC, similar to the findings for pomelo fruits [45].

The antioxidant capacity of DPPH scavenging assay was chosen to determine the total antioxidant capacity of different tissues of the two sweet oranges (Table 2). Like total phenolics and flavonoids, flavedo and albedo ranked higher for DPPH scavenging activity ($\geq$65%) than all other fruit tissues, regardless of the variety. These findings were also confirmed by Rahman et al. [28] that found a much higher amount of DPPH scavenging activity in the flavedo rather than in all other fresh fruit tissues. Juice sacs of Valencia Late contained a significant higher DPPH scavenging activity than the ones of Natal IAC, showing that the same fruit tissue may have some variation on the expression of this activity depending on the species. In general, obtained by-product of both varieties, recognized as residues of the juice processing industry, contain excellent antioxidant capacity and may be used in food products as active ingredients or substitutes for synthetic preservatives [46], preventing losses along the food chain.

### 4.2. Postharvest Fruit Quality Evaluations

The major role of coating materials on fruit surfaces is to reduce water losses and to extend the fruit shelf life [14]. Even minor losses of water, between a 5 and 6% range, can affect the juice content, appearance and firmness of the citrus fruit resulting in economic

losses due to the reduction in the saleable weight [1]. In our study, the weight losses were influenced by the length of the storage period and the wax-coating treatments but showed similar tendency for the two sweet orange varieties (Figure 2). These results suggest that both late-season varieties do not only share similarities in their physicochemical properties (Table 1), but also on their postharvest quality attributes (Figure 2). A significant ($p \leq 0.001$) progressive increase in weight losses was observed over the storage period for both varieties and wax-coating treatment. Uncoated fruits had the highest weight losses during the entire storage period. However, the wax-coating treated fruits had weight losses below 15% after 60 days of cold storage, which supports the findings reported for Salustiana and Champagne sweet oranges stored for 60 days at $3 \pm 1\,^{\circ}C$ and 80–90% RH [33,47]. Similarly, Pereira et al. [48] found significant variation in weight losses after 28 days of storage at $24 \pm 2\,^{\circ}C$ and 35–45% RH for Valencia Delta sweet orange, comparing non-coating with carnauba wax treated fruit. These authors noticed that Valencia Delta sweet orange fruits receiving wax-coating treatment lost around 14% of their initial weight compared to 26% for the fruits without any coating treatment. Lower weight loss rates was also reported for Navelina and Washington navel oranges after 60 days of cold storage at $3 \pm 1\,^{\circ}C$ and 80–95% RH, with a range of 1 to 5% of weight losses [49,50]. These results indicate that each citrus variety has different postharvest behavior depending on the postharvest handling and the length of the storage period. Furthermore, in previous work, we suggested that these variations may be resulted from the genetic attributes of each variety associated with the storage conditions involving temperature, humidity, air movement and atmospheric gases [33].

The water losses are detrimental to the physicochemical and organoleptic quality of the fruit. This is a major issue for the postharvest process and commercialization, particularly for citrus fruits that contain around 80–90% water [1]. Our results demonstrated that fruits without any wax-coating treatment had an increase in weight loss by 67% compared to carnauba wax/wood resin treatment following 60 days of cold storage. A lack of wax-coating treatments promotes fruit decay, including firmness loss and depreciation due to intense metabolic rate and water losses [33].

Fruit color directly influences consumer's perception and acceptance [51]. The color development was quantitatively expressed as citrus color index, CCI (Figure 4). CCI became an accurate and a reliable parameter to measure color changes in citrus fruit and was widely applied by the citrus industry to establish harvest time and postharvest treatments [26]. The fresh fruit market requires sweet orange fruits with attractive peel color and without any damage [25,37]. In our study, the uncoated fruits (control) exhibited a progressive color evolution in the flavedo, for both varieties (Figure 3). In contrast, no color development was observed in the peel of the wax-coated fruits. This indicates that the wax-coating treatment may have inhibited the chlorophyll degradation [26] and blocked the pathway of carotenoid synthesis [52] by reducing the metabolic rate and ethylene biosynthesis in the fruits. In most cases, coating delays the natural senescence process and the biosynthesis of colored pigments in citrus, resulting in a lagging color change [20]. Interestingly, the excessive water losses observed for uncoated fruits (Figure 2) may have favored the ethylene biosynthesis that led to the chlorophyll degradation and color change, as reported by Nasrin et al. [20].

Similar to the external appearance, the internal quality of the sweet orange fruits plays an important role on consumer's acceptance and marketability [1]. Therefore, sweet orange fruits must achieve certain quality standards before commercialization [25,37]. Significant differences were observed among the tested postharvest treatments for TSS content (Table 3). TSS concentration ranged from 10.4 up to 12.1 °Brix. The lowest TSS content was found for Natal IAC without any coating treatment. However, fruits of both varieties attended the minimal TSS value required by the fresh citrus market, 10 °Brix [25,53]. No such variation was observed among the treatments for the acidity level. The citric acid content ranged between 0.5 and 1.0 g·100 mL$^{-1}$, which is considered an optimal range to confer proper organoleptic quality to sweet oranges [39]. Natal IAC fruits combined with wax-coating had the lowest sugar-acidity ratio, 15:1, while the non-coated Valencia Late fruits ranked

higher level for this variable, above 18:1. All other treatments had intermediate values for sugar/acidity ratio. These indices indicate a suitable fruit maturity for both sweet oranges and postharvest treatments, as the ratios were all above the minimal standard required by the fresh fruit market [25,37]. Sugar/acidity ratio may exceed 20:1 after harvest, depending on variety and postharvest treatment, but the citric acid level cannot be below $0.4 \text{ g·}100 \text{ mL}^{-1}$, so that the sweet orange fruit do not taste too insipid with much less acidity, becoming unsuitable for fresh consumption [1,54]. Interestingly, the length of the storage period, up to 60 days, did not significantly affect the internal quality of the fruit of both varieties, regardless of the postharvest treatment. This is important, as the internal quality of the fruit was preserved across the evaluated period. Moreover, the postharvest treatment with carnauba wax/wood resin did not cause any depreciation or decay of the fruit after 60 days of cold storage at $5 \pm 1$ °C and 60–70% RH.

Juice content is an important characteristic that modulates fruit quality in sweet oranges. The use of this parameter as a commercial standard index is dependent on the variety and the market destination [51]. According to the international fresh fruit market, sweet oranges must have at least 35% of juice content [1,37]. Based on this requirement, the fruit of both varieties had appropriate juice content in the first two periods of storage. However, at 30, 45 and 60 days of cold storage, the uncoated fruit had much less juice content than the minimum required by the fresh market [1,37]. These results agree with previous studies that showed a progressive reduction in the juice content of non-treated fruit by increasing the length of the storage [1,50]. The juice content of Natal IAC fruits treated with carnauba wax and wood resin did not change during the storage period (Figure 4). This may be due to the moderate water loss observed for the wax-coated fruits (Figure 2). Furthermore, the morphological traits of this variety may have contributed to adjust the juice content, as the fruits of Natal IAC were moderately smaller than Valencia Late (Table 1). This may result in a low fruit surface–area for water loss by transpiration. Davies and Zalman [55] pointed out that larger orange fruits have proportionally lower juice content than smaller ones, supporting our findings.

The progression of the fruit color was also observed by the panelist group in the sensory analysis of uncoated fruits (Figure 5). Panelists expressed higher preference for wax-coated fruits at the beginning of the postharvest, by 30 days of cold storage, probably due to the fruit glossiness that improved the attractiveness. Thereafter, uncoated fruits ranked high for this attribute, as they exhibited a deeper yellow color, which was confirmed by the color measurements (Figure 3). It is worth mentioning that fruit color is an important factor to determine purchase intent for fresh citrus [1]. This evidence was reported in our study, as the panelists expressed their preference according to fruit attractiveness involving glossiness and improved color.

Regarding fruit firmness, the panelists had less preference for uncoated fruits as storage progressed, particularly for Valencia Late, which was also confirmed for fruit texture. The lower preference for uncoated fruits based on firmness was probably due to the excessive water losses (Figure 2), as the cell vacuole shrinks and the turgor potential declined to zero when fruit loss water, favoring cell collapse and consequent fruit shrinkage inferring on panelist preference [34]. The rates for aroma and taste attributes remained constant up to 60 days of storage period. This agrees with the results found for juice quality in which no significant variations were observed (Table 3). However, the panelists indicated higher preference for wax-coated than uncoated fruits according to the sensory profile after 60 days of cold storage, demonstrating the effectiveness of the carnauba wax/wood resin in preserving fruit quality after a long storage period. The juiciness of the fruit was not differentially perceived by the panelist group over the storage period. However, the panelist expressed much higher preference for Valencia Late combined with wax-coating treatment after 60 days of storage, reinforcing the efficiency of this postharvest treatment. Taken together, carnauba wax and wood resin associated with the cold storage brought beneficial results to the citrus fruit postharvest conservation involving the delay of water

loss, natural senescence and fruit color development by the maintenance of physicochemical and sensorial quality attributes.

## 5. Conclusions

Fruits of Valencia Late and Natal IAC sweet oranges showed excellent physicochemical quality and antioxidant capacity, attending the requirements of the fresh citrus market. Flavedo and albedo sections of both studied sweet oranges contained high concentrations of phenolics and flavonoids, as well as the antioxidant activity. Therefore, these fruit tissues, recognized as residues left from the citrus processing industry, may be potentially used in food products as a remarkable low-cost antioxidant source.

Carnauba wax/wood resin coating treatment combined with cold storage at $5 \pm 1$ °C and 60–70% RH was efficient to delay fruit color development and to prevent weight losses for both sweet oranges up to 60 days. Moreover, the physicochemical and sensory quality of wax-coated fruits were preserved during cold storage, while uncoated fruits showed a decrease in juice content and lower preference as the storage period was prolonged for up to 60 days. Overall, the effect of carnauba wax and wood resin in prolonging shelf-life of Valencia Late and Natal IAC fruits is positive by reducing weight loss, inhibiting senescence and color progression, as well as preserving the physicochemical and organoleptic quality of the fruit.

**Author Contributions:** Conceptualization, R.P.L.J. and D.U.d.C.; methodology, D.U.d.C. and F.A.; formal analysis and investigation, D.U.d.C., M.A.d.C. and R.C.C.; data curation and writing—original draft preparation, D.U.d.C.; writing—review and editing, R.P.L.J., C.S.V.J.N. and F.A.; supervision and funding acquisition, R.P.L.J. and F.A. All authors have read and agreed to the published version of the manuscript.

**Funding:** This research was partially supported by the Coordenação de Aperfeiçoamento de Pessoal de Nível Superior (Capes, finance code 001), Universidade Estadual de Londrina (UEL), Instituto de Desenvolvimento Rural do Paraná—IAPAR/Emater (IDR-Paraná) and by the USDA National Institute of Food and Agriculture (NIFA) Hatch project (Project number 1019171).

**Institutional Review Board Statement:** The study was conducted in accordance with the Declaration of Helsinki.

**Informed Consent Statement:** Informed consent was obtained from all subjects involved in the study.

**Data Availability Statement:** All data generated and analyzed during this study are presented in the published version of this article.

**Acknowledgments:** D.U.d.C. acknowledges the Coordenação de Aperfeiçoamento de Pessoal de Nível Superior (Capes) by providing the Ph.D. scholarship (Grant number 88881.361958/2019-01).

**Conflicts of Interest:** The authors declare no conflict of interest.

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
