# Peer review of "Effectiveness of Natural-Based Coatings on Sweet Oranges Post-Harvest Life and Antioxidant Capacity of Obtained By-Products"

_horticulturae, doi:10.3390/horticulturae9060635_

Round 1

Reviewer 1 Report

The manuscript was to investigate the effect of coatings in prolonging sweet orange fruit post-harvest life and quality evaluation of by-products. The topic is of some interesting since it is about the application of industry by-products. The authors can address the points below to revise your manuscript.

General comments,

1. The English language can be checked again some minor points should be revised.

2. In the introduction, please update the relevant references to emphasize the importance of your study. 

3. In the methods, when it comes to the evaluation of the juice quality. The reviewer suggests to add some experiments. For example, please discuss the change of the total phenolic contents and antioxidant activity, as well as the components analysis such as the dietary fiber contents in coating/non-coating samples.

4. In the discussion section, please provide potential mechanism of the effect of natural-based coatings on the quality.

The English language can be checked again some minor points should be revised.

Author Response

Dear Reviewer,

We would like to thank you for all the suggestion that you have done to improve the quality of our MS. All the significant changes were incorporated in the latest version of our MS.

  1. The English language can be checked again some minor points should be revised.

Answer: The MS was revised by a language service.

  1. In the introduction, please update the relevant references to emphasize the importance of your study.

Answer: We have emphasized the importance of our study in the introduction of our MS with the most relevant references.

  1. In the methods, when it comes to the evaluation of the juice quality. The reviewer suggests to add some experiments. For example, please discuss the change of the total phenolic contents and antioxidant activity, as well as the components analysis such as the dietary fiber contents in coating/non-coating samples.

Answer: The total phenolic content and antioxidant activity was performed in our study and properly discussed in discussion section. However, we did not run any analysis of dietary fiber, for that reason it was not debated in our MS.

  1. In the discussion section, please provide potential mechanism of the effect of natural-based coatings on the quality.

Answer: The effect of natural-based coating on the quality of the fruit was incorporated in the text.

Reviewer 2 Report

These are some comments:

1.    In Figure 3, the control fruit reached weight loss of 25% at 60 days of storage. How much weight loss did the sweet orange fruit reach to lose commercial value?

2.   
Do the data in Table 2. indicate no change in TSS and TA levels during storage of the fruit? Is this result reasonable?

3.   
The data in Table 2. is not very easy to interpret, suggest changing it to a more understandable form.

4.    What are the possible reasons why the treatment effectively reduced the weight loss of Valencia Late sweet orange fruit, but its juice content continued to decrease as in the control?

5.    The method does not describe whether the juice is made from whole fruit or peeled juice? Does the treatment have an effect on the total phenols and flavonoids of the remaining residue, and what kind of effect is it?

Minor editing of the English language is required to increase the readability of the article

Author Response

Dear Reviewer,

We would like to thank you for all the suggestion that you have done to improve the quality of our MS. All the significant changes were incorporated in the latest version of our MS.

The English language can be checked again some minor points should be revised.

Answer: The MS was revised by a language service.

These are some comments:

     1. In Figure 3, the control fruit reached weight loss of 25% at 60 days of storage. How much weight loss did the sweet orange fruit reach to lose commercial value?

Answer: Even minor losses of water, between a 5–6% range, can affect the juice content, appearance and firmness of citrus fruit resulting in economic losses due to the reduction of the saleable weight. However, variation on water loss can be observed among the species and varieties.

     2. Do the data in Table 2. indicate no change in TSS and TA levels during storage of the fruit? Is this result reasonable?

Answer: Yes, we did not find significant (P < 0.05) differences between the tested treatments over the storage period (days). Moreover, no significant interaction among the factors treatments and storage periods were observed.  

     3. The data in Table 2. is not very easy to interpret, suggest changing it to a more understandable form.

Answer: The mean data of each singular factor (postharvest treatments and storage period) were presented in this way (table format) because no significant interaction among the studied factors was found for the variables total soluble solids (TSS), titratable acidity (TA) and TSS/TA ratio.   

     4. What are the possible reasons why the treatment effectively reduced the weight loss of Valencia Late sweet orange fruit, but its juice content continued to decrease as in the control?

Answer: We suggest that the juice content in Valencia Late fruit decreased as the length of the storage period was extended probably because of its larger surface area compared to Natal IAC fruit, with a more intense loss of water and juice content. However, this difference in water loss did not cause any effect on fruit internal quality as observed in the results.

     5. The method does not describe whether the juice is made from whole fruit or peeled juice? Does the treatment have an effect on the total phenols and flavonoids of the remaining residue, and what kind of effect is it?

Answer: The juice was made from peeled fruit.

Reviewer 3 Report

I reviewed the manuscript entitled Effectiveness of Natural-based Coatings in Prolonging Sweet Orange Fruit Post-Harvest Life and Evaluation of the Antioxidant Capacity of Obtained By-products.

The authors were reported the effectiveness of the carnauba wax/wood resin based coating to prolong the postharvest life under cold storage and the physicochemical quality/antioxidant capacity of the late-season Valencia Late and Natal IAC sweet oranges fruit and their by-products (peel, pulp, and rag). This extension of fruit shelf life and waste utilization has great application value.

I agree to accept this manuscript, but there are a few issues that need to be revised before acceptance. 

1) The article needs to be polished by a native English expert.

2) Table 1. and Table 2 a, b, ns should be superscript.

3) Figure 3 and Figure 5 a, b, and ab, which represent differences, should be superscripts. Does p 0.001 in the figure refer to 60 days?

4) In the references, the year font should be bold. For example, ref 1-3, 5, 13, 22, 24-25, 31, 36-39, 53.

5) Ref 8, Ind in the magazine name should also be italicized.

6) Some book titles are italicized, such as ref 1, 3, 13, 22, 24-25, 27, 36. But ref 31, 38-39 do not have italics, please unify them.

The article needs to be polished by a native English expert.

Author Response

Dear Reviewer,

We would like to thank you for all the suggestion that you have done to improve the quality of our MS. All the significant changes were incorporated in the latest version of our MS.

1) The article needs to be polished by a native English expert.

Answer: The MS was revised by a language service.

2) Table 1. and Table 2 a, b, ns should be superscript.

Answer: We changed it.

3) Figure 3 and Figure 5 a, b, and ab, which represent differences, should be superscripts. Does p ≤ 0.001 in the figure refer to 60 days?

Answer: The letters that indicate the differences between treatments (variety and wax-coating) were changed to superscripts. The p ≤ 0.001 indicates the level of significance in the ANOVA.

4) In the references, the year font should be bold. For example, ref 1-3, 5, 13, 22, 24-25, 31, 36-39, 53.

Answer: They were corrected.

5) Ref 8, Ind in the magazine name should also be italicized.

Answer: It was corrected.

6) Some book titles are italicized, such as ref 1, 3, 13, 22, 24-25, 27, 36. But ref 31, 38-39 do not have italics, please unify them.

Answer: They are italicized in the text.

Round 2

Reviewer 1 Report

The authors have answered the reviewers' questions.

Author Response

Dear Reviewer,

First, we would like to thank you for all the comments and suggestions. Based on your comments, we have edited and revised the text to improve the quality of our manuscript.